# Mechanical Reliability Assessment by Ensemble Learning

**Weizhen You [1], Alexandre Saidi [2], Abdel-malek Zine [3] and Mohamed Ichchou [1,\*]**

[1]  LTDS, Ecole Centrale de Lyon, 69134 Écully, France; weizhen.you@ec-lyon.fr
[2]  LIRIS, Ecole Centrale de Lyon, 69134 Écully, France; alexandre.saidi@ec-lyon.fr
[3]  ICJ, Ecole Centrale de Lyon, 69134 Écully, France; abdel-malek.zine@ec-lyon.fr
\*  Correspondence: mohamed.ichchou@ec-lyon.fr

**Abstract:** Reliability assessment plays a significant role in mechanical design and improvement processes. Uncertainties in structural properties as well as those in the stochatic excitations have made reliability analysis more difficult to apply. In fact, reliability evaluations involve estimations of the so-called conditional failure probability (CFP) that can be seen as a regression problem taking the structural uncertainties as input and the CFPs as output. As powerful ensemble learning methods in a machine learning (ML) domain, random forest (RF), and its variants Gradient boosting (GB), Extra-trees (ETs) always show good performance in handling non-parametric regressions. However, no systematic studies of such methods in mechanical reliability are found in the current published research. Another more complex ensemble method, i.e., Stacking (Stacked Generalization), tries to build the regression model hierarchically, resulting in a meta-learner induced from various base learners. This research aims to build a framework that integrates ensemble learning theories in mechanical reliability estimations and explore their performances on different complexities of structures. In numerical simulations, the proposed methods are tested based on different ensemble models and their performances are compared and analyzed from different perspectives. The simulation results show that, with much less analysis of structural samples, the ensemble learning methods achieve highly comparable estimations with those by direct Monte Carlo simulation (MCS).

**Keywords:** structural reliability; uncertainties; ensemble learning

## 1. Introduction

Reliability describes the probability that the object realizes its functions under given conditions for a specified time period [1]. As a way to improve the quality of products, reliability assessment is carried out by companies such that they can make product planning and implement preventive maintenance. For a mechanical structure subjected to stochastic excitation, an important task is to evaluate the risk of structure failures, in other words, the failure probability of the structure. Mathematically, the failure probability is determined by a multi-dimension integral over the spaces of all possible variables (factors) involved, i.e., [2]

$$P_f = \int_{g(\mathbf{z}) \leq 0} p_Z(\mathbf{z}) d\mathbf{z} = \int_{\mathbf{z} \in \Omega_Z} I_f(\mathbf{z}) p_Z(\mathbf{z}) d\mathbf{z} = E_{\mathbf{z} \in \Omega_Z}[I_f(\mathbf{z})], \tag{1}$$

where $\mathbf{z}$ is a vector that consists of the variables involved in the excitation. $g(\mathbf{z})$ is a defined performance function used to identify structural failures, that is to say, $g(\mathbf{z}) \leq 0$ when $\mathbf{z}$ falls into failure region. $\Omega_Z$ is the uncertainty space of $\mathbf{z}$; $I_f(\mathbf{z})$ is an indicator function that equals 1 when $g(\mathbf{z}) \leq 0$ and 0, otherwise; $E[\cdot]$ denotes the mathematical expectation. In fact, due to various factors (manufacturing, environment, fatigue . . . ), the structural properties become uncertain. Hence, the assessment of failure

probability involves the uncertainties in both structural parameters and the excitation. Considering the relative independence between the uncertainties of the structural properties and those of the excitation, the conditional failure probability can be defined and formulated as [3]

$$P_f^c(\mathbf{x}) = \int_{g(\mathbf{x},\mathbf{z})\leq 0} p_Z(\mathbf{z})d\mathbf{z} = \int_{\mathbf{z}\in\Omega_Z} I_f(\mathbf{z}|\mathbf{x})p_Z(\mathbf{z})d\mathbf{z} = E_{\mathbf{z}\in\Omega_Z}[I_f(\mathbf{z}|\mathbf{x})], \tag{2}$$

where $\mathbf{x}$ is a vector that consists of the uncertain properties of the object structure. $I_f(\mathbf{z}|\mathbf{x})$ is a 'conditional' indicator function of $\mathbf{z}$ in terms of $\mathbf{x}$. Generally, in the analysis of structural dynamic responses, the stochastic excitation process is discretized so that a discrete response process is achieved. Then, the characterization of the integrals in Equations (1) and (2) may involve hundreds of random variables in the context of stochastic loading. Therefore, the current reliability problem forms actually a high dimensional problem.

Compared with the dimension of structural properties, the dimension of the excitation is always very high. Direct MCS demands a huge number of samples to ensure a high accuracy of the estimated CFP, but this will be very time-consuming. In this aspect, surrogate models become a good alternative. Surrogate models are developed to deal with highly nonlinear or implicit performance functions. They are introduced to reduce computational burden in reliability analysis. Assuming $\mathbf{x} = (x_1, x_2, \ldots, x_n)$ is a $n$-dimension input vector, $y$ is the output. For a data set with $N$ samples, $\mathbf{X} = (\mathbf{x}^1; \mathbf{x}^2; \ldots; \mathbf{x}^N)$, the corresponding responses are $\mathbf{Y} = (y^1, y^2, \ldots, y^N)^T$. Assume that the function between the input variables and output response takes the following form $y(\mathbf{x}) = \hat{y}(\mathbf{x}) + \varepsilon$, where $y(\mathbf{x})$ is the unknown response function, $\hat{y}(\mathbf{x})$ is a certain surrogate function seen as an approximation of the real function. $\varepsilon$ is the error induced by the surrogate function. A surrogate model technique mainly consists of three parts. Firstly, a number of samples are taken from the input space $\mathbf{X}$; secondly, the output responses $\mathbf{Y}$ of these samples are determined through numerical models such as finite element analysis (FEA), resulting in a training data set $[\mathbf{X}_{N\times n}, \mathbf{Y}_{N\times 1}]$ for the surrogate model; finally, the training data are employed to train a surrogate model. See the illustration in Figure 1.

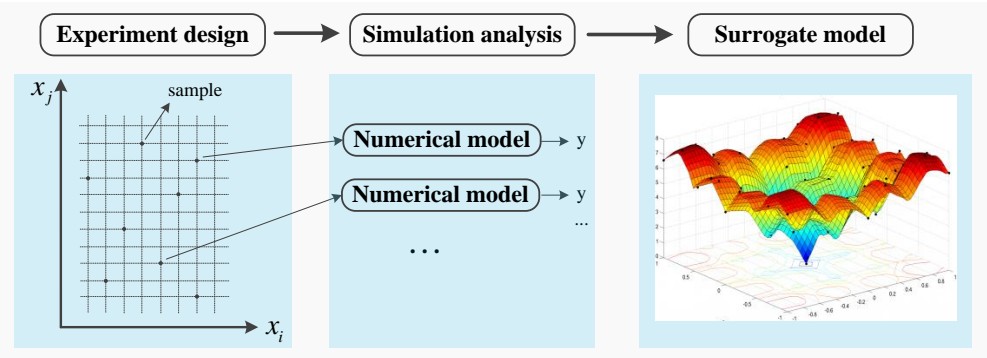

**Figure 1.** Theoretical framework to estimate failure probability based on an ML model.

Response surface method (RSM) [4] is among the most popular surrogate models. Some researchers combine RSM with other approaches to refine the model parameters so that the model becomes more efficient [5]. Support vector machine (SVM) has also gained much attention recently. Pan and Dias [6] combined an adaptive SVM and MCS to solve nonlinear and high-dimensional problems in reliability analysis. Other surrogate models such as metamodel, ANNs, and Kriging have been proposed by other researchers [7–9]. However, they cannot avoid shortcomings in all situations. The disadvantages in ANNs mainly include the complex architecture optimization, low robustness, and enormous training time [10]. SVM is time-consuming for large-scale applications and sometimes shows a large error in sensitivity calculation of reliability index. RSM has been popular; however, it may be time-consuming to use the polynomial function when the components are complex and the number of random variables is

large [11]. Moreover, the approximate performance function is lack of adaptivity and flexibility, and we cannot guarantee that it is sufficiently accurate for the true one.

In the authors' viewpoint, the evaluation of CFP in Equation (3) can be seen as a regression problem that takes a realization of the structural uncertain properties as input and the CFP as output. To improve reliability assessment considering structural uncertainties, more attention should be paid to the non-parametric statistical learning methods, such as RF [12] and GB [13] et al. According to the current research state, we believe that the explorations of ML methods are far from enough. The rest of the paper is organized as follows. In Section 2, the theories pertaining to CFP and its estimation are presented. In Section 3, the framework of failure probability estimation based on ML models is introduced. In Section 4, the principles of ensemble learning methods are presented. In Section 5, numerical simulations on different structures are applied, and discussion of the results are made. Section 6 makes some concluding remarks.

## 2. Failure Probability Estimation: ML-Based Framework

As mentioned before, the calculation of the overall failure probability considers both uncertainties of the stochastic excitation and those of the structural properties. In this aspect, the target failure probability is estimated according to the formula below:

$$
\begin{aligned}
P_f &= \int_{g(\mathbf{x},\mathbf{z})\leq 0} p_X(\mathbf{x}) p_Z(\mathbf{z}) d\mathbf{x} d\mathbf{z} \\
&= \int_{\mathbf{x}\in\Omega_X, \mathbf{z}\in\Omega_Z} I_f(\mathbf{x},\mathbf{z}) p_X(\mathbf{x}) p_Z(\mathbf{z}) d\mathbf{x} d\mathbf{z} \\
&= \int_{\mathbf{x}\in\Omega_X} \{ \int_{\mathbf{z}\in\Omega_Z} I_f(\mathbf{z}|\mathbf{x}) p_Z(\mathbf{z}) d\mathbf{z} \} p_X(\mathbf{x}) d\mathbf{x} \\
&= \int_{\mathbf{x}\in\Omega_X} P_f^c(\mathbf{x}) p_X(\mathbf{x}) d\mathbf{x} \\
&= E_{\mathbf{x}\in\Omega_X}[P_f^c(\mathbf{x})],
\end{aligned}
\tag{3}
$$

which means that the target failure probability can be seen as the expectation of CFP over the uncertainty space of structural properties. From an ML perspective, training data are firstly prepared to train a model, then this model is employed to make predictions on a new data. The training data consist of several samples of the input vector and the output values while the new data only contain the samples of the input vector. When the CFPs are available for a few samples of the uncertain structural properties, an ML model is trained that takes the samples of **x** as inputs and the corresponding CFPs as outputs. Then, this model is employed to predict the outputs on the new data. Machine learning is a very powerful tool to find potential relationships within the data and make predictions. In this research, only a few hundred samples are needed to train an accurate ML model, then the model can be used to make immediate predictions on the new samples.

Figure 2 shows the framework of the proposed method. Firstly, a set of random samples are generated from the structural uncertainty space; secondly, the samples are split into two subsets, one for training (10%) and the other (90%) for making predictions; thirdly, MCS is employed to estimate target values (CFPs) of the training set; fourthly, an ML model is trained from the training data; fifthly, the ML model is used to make predictions on the prediction set. Finally, the predictions are averaged to obtain the estimation of overall failure probability. It is noticed that the pre-processing of the random samples of the structure is necessary. In this research, the samples are adjusted so that they follow truncated distributions. In Figure 2, the samples to train the ML model are different from the samples for making predictions.

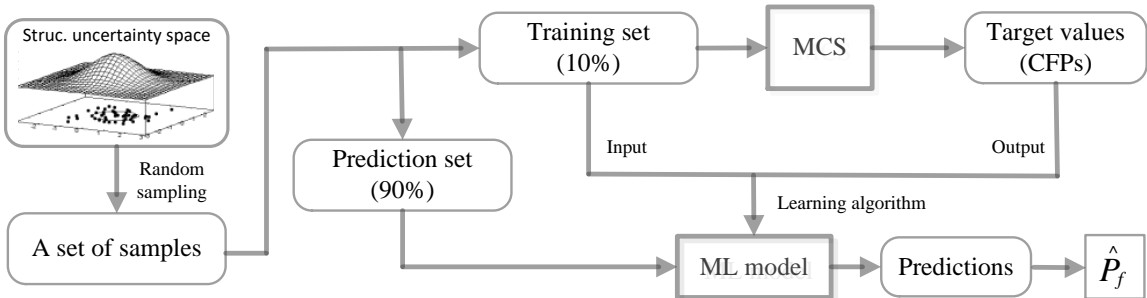

**Figure 2.** Theoretical framework to estimate failure probability based on the ML model.

## 3. MCS to Prepare the Training Data

### 3.1. General MCS to Estimate CFPs

In the perspective of first-excursion probability [2], it is general to compare the responses of interest $y_i(t), i = 1, \ldots, n_r$ against acceptable threshold levels $y_i^*$ within the time duration $T$ of the stochastic excitation. A failure takes place whenever the response $y_i$ exceeds its corresponding threshold. From this point of view, failure implies not meeting the predefined conditions (this does not necessarily imply collapse). In terms of performance function, the failure event $F$ is defined as:

$$F = \{\mathbf{x} \in \Omega_X, \mathbf{z} \in \Omega_Z : g(\mathbf{x}, \mathbf{z}) \leq 0\} \tag{4}$$

The value of $g(\cdot)$ is equal to or smaller than zero whenever a response exceeds its prescribed threshold. Considering the different DOFs of the structure as well as the sampled time points, the performance function $g(\mathbf{x}, \mathbf{z})$ is formulated as [2]

$$g(\mathbf{x}, \mathbf{z}) = 1 - \max_{i=1,\ldots,n_r} \left( \max_{k=1,\ldots,n_T} \left( \frac{|y_i(t_k, \mathbf{x}, \mathbf{z})|}{y_i^*} \right) \right), \tag{5}$$

where $|\cdot|$ denotes absolute value; $t_k$ is the $k$th time point within the time interval $[0, T]$; $y_i$ is the response of $i$th degree of freedom (DOF) of the structure; $y_i^*$ is a defined threshold for the $i$th DOF. By direct MCS, the $P_f^c(x)$ can be approximated as

$$P_f^c(\mathbf{x}) = E[I_f(\mathbf{z}|\mathbf{x})] = \frac{1}{N} \sum_{i=1}^{N} I(g(\mathbf{z}_i|\mathbf{x}) < 0), \tag{6}$$

where $I(\cdot) = 1$ if $g(\cdot) < 0$ and 0, otherwise. The $N$ realizations of the stochastic excitation, $\mathbf{z}_1, \mathbf{z}_2, \ldots, \mathbf{z}_N$, are treated as individual inputs of a finite element code that describes the structural responses.

As already known, the estimations by direct MCS can be treated as a standard reference. However, it is not computationally efficient for estimating very small failure probabilities (for example, <1%) since the number of samples required to achieve a given accuracy is inversely proportional to the scale of $P_f$. In other words, estimating small probabilities requires information from rare samples that induce structural failures. On average, it requires many samples before one such failure sample occurs. To ensure a demanded accuracy of the estimation, the number of samples needed to be analyzed is $n = (1/P_f - 1)/c^2$ [14], where $P_f$ is the actual failure probability, and $c$ is the coefficient of variation. For example, if $c = 0.1$ and the actual $P_f$ is $10^{-1}$, then only $(1/10^{-1} - 1)/0.1^2 = 9 * 10^2$ samples need to be analyzed. However, if the actual $P_f$ value is $10^{-3}$, then at least $(1/10^{-3} - 1)/0.1^2 = 9.99 * 10^4$ samples are needed, which consumes much more CPU time. Therefore, the standard MCS is only practically suitable to calculate relatively larger $P_f$ (for example, >1%).

In view of this, the importance sampling (IS) method is a good choice to compute small $P_f$ values. In principle, an IS method tries to adjust the sampling density so that more samples from the failure region $F$ can be obtained. The efficiency of the method relies on the construction of the importance sampling density (ISD), for which the knowledge about the failure region is inevitably required.

### 3.2. Importance Sampling to Estimate Very Small CFPs

Consider a linear structural system represented by an appropriate model (e.g., a finite element model) comprising a total of $n$ DOFs. The system is subjected to a Gaussian process excitation with zero mean, $z(t)$. Then, the system response can be evaluated by the convolution integral [15],

$$y_i(t, \mathbf{x}) = \int_0^t h_i(t - \tau, \mathbf{x}) z(\tau) d\tau, \tag{7}$$

where $h_i(t - \tau, \mathbf{x})$ is the unit impulse response function for the $i$th DOF of the structure at time $t$ due to a unit impulse applied at time $\tau$. $\mathbf{x}$ is a vector that consists of the structural properties. Generally, the zero initial condition at $t = 0$ is assumed. As a result of linearity, the response of interest is actually a linear combination of the contributions from each input $z(\tau)$. Considering that the excitation is modeled as a Gaussian process, its discrete form can be approximated as K-L expansions [16], i.e.,

$$z(t) = \mu(t) + \sum_{i=1}^M \sqrt{\lambda_i} u_i \phi_i(t), \tag{8}$$

where $\mu(t)$ is the mean function and $u_i, i \in N^+$ are standard normal variables. $M$ is the number of truncated terms to keep. The eigenpair $\{\lambda_i, \phi_i\}$ is the solution of the eigenvalue problem $\int_B C_{HH}(x, x') \phi_i(x') dx' = \lambda_i \phi_i$, which is a Fredholm integral equation of the second kind. The kernel $C_{HH}(x, x')$, being an autocovariance function, is symmetric and positive definite. The set of eigenvalues is, moreover, real, positive, and numerable. By Equations (7) and (8), we obtain

$$y_i(t, \mathbf{x}) = \int_0^t h_i(t - \tau, \mathbf{x}) \sum_{l=1}^M u_l f_l^{KL}(\tau) d\tau = \sum_{l=1}^M u_l a_l(t, \mathbf{x}) = \mathbf{a}(t, \mathbf{x})^T \mathbf{u}, \tag{9}$$

where $f_l^{KL}(t) = \sqrt{\lambda_l} \phi_l(t)$, and $\mathbf{a}(t, \mathbf{x}) = [a_1(t, \mathbf{x}), \dots, a_M(t, \mathbf{x})]^T$, where

$$a_l(t, \mathbf{x}) = \int_0^t h_i(t - \tau, \mathbf{x}) f_l^{KL}(\tau) d\tau, l = 1, \dots, M \tag{10}$$

From Equation (9), the response process is found to be the scalar product of two vectors: the random vector $\mathbf{u}$ and the deterministic basis function vector $\mathbf{a}(t, \mathbf{x})$, whose elements are convolutions of the basis functions $f_l^{KL}(t)$ of the excitation and the unit impulse response function $h_i(t - \tau, \theta)$ of the system.

Now, consider the excitation that leads to the event $y_i(t_j) \geq y_i^*$ at time $t = t_j$, where $y_i^*$ is a designed threshold for the $i$th DOF. This corresponds to realizations of $\mathbf{u}$ that satisfy the condition

$$y_i^* - |\mathbf{a}_i(t_j, \mathbf{x})^T \mathbf{u}| \leq 0 \tag{11}$$

In the space of $\mathbf{u}$, these realizations lie in a pair of symmetric subspaces bounded by the hyper-planes $y_i^* = \pm \mathbf{a}_i(t_j, \mathbf{x})^T \mathbf{u}$ that have the same distances $\beta_i(t_j, \mathbf{x}) = y_i^* / ||\mathbf{a}_i(t_j, \mathbf{x})||$ from the origin. In addition, $g_i(\mathbf{u}) = y_i^* - |\mathbf{a}_i(t_j, \mathbf{x})^T \mathbf{u}|$ is the performance function, $g_i(\mathbf{u}) = y_i^* - |\mathbf{a}_i(t_j, \mathbf{x})^T \mathbf{u}| = 0$ is the limit-state surface; $\beta_i^+(t_j, \mathbf{x}) = \beta_i^-(t_j, \mathbf{x}) = \beta_i(t_j, \mathbf{x})$ are the reliability indices for the event $g_i(\mathbf{u}) \leq 0$ at time $t_j$. The failure probability of the $i$th DOF at time $t_j$ is denoted as [15]

$$P(g_i(\mathbf{u}) \leq 0) = 2 * \Phi(-\beta_i(t_j, \mathbf{x})), \tag{12}$$

where $\Phi(.)$ denotes the standard normal cumulative probability function. Figure 3 provides an illustration of an elementary failure region $F_{i,j}(\mathbf{x})$ with $n_r = 1$, $n_T = 1$, $n_{KL} = 2$.

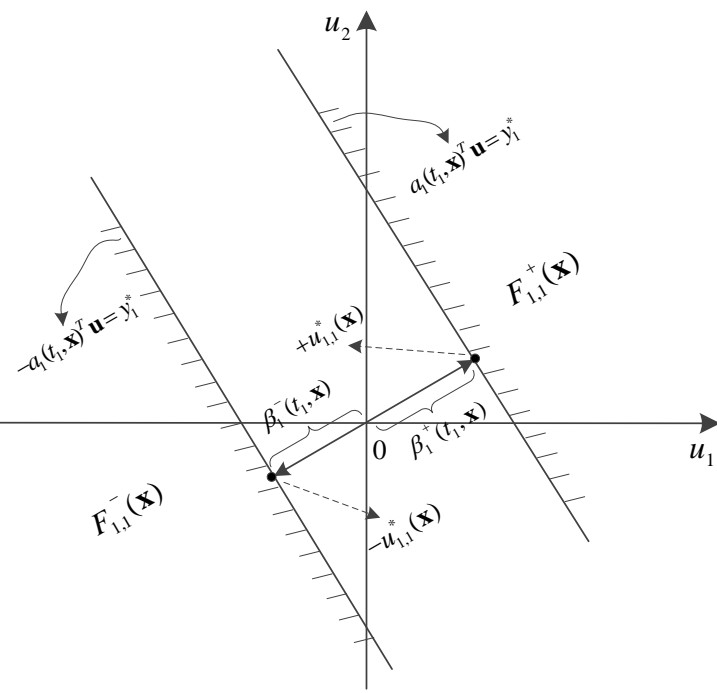

**Figure 3.** Illustration of a symmetric elementary failure region [3].

To apply the IS technique to evaluate the CFP, the integral in Equation (2) is re-written as

$$\hat{P}_f^c(\mathbf{x}) = \int_{u \in \Omega_u} I_F(u|\mathbf{x}) \frac{p_u(u)}{p_{IS,u}(u)} p_{IS,u}(u) du = E_{u \in \Omega_u}[I_f(u^v|\mathbf{x}) \frac{p_u(u^{(v)})}{p_{IS,u}(u^{(v)})}] \tag{13}$$

where $p_{IS,u}(u)$ is the Importance Sampling density (ISD) function and $u^{(v)}, v = 1, \ldots, and N$ are samples of uncertain vector $U$ obtained via the ISD $p_{IS,u}(u)$. The most important part for implementing the IS procedures is the design of the ISD function that is able to obtain more samples in the failure region, meanwhile ensuring a low variability of the estimated probability, i.e., failure samples are drawn frequently while the variability of the ratio involving $p_u(u)$ and $p_{IS,u}(u)$ is low. Based on the elementary failure regions, a well defined ISD function is employed in this paper. See more details in Au and Beck [2].

## 4. Train the ML Model by Ensemble Learning

The ensemble learning methods mainly include Bagging, Boosting, and Stacking. In principle, Bagging adopts Bootstrap sampling to learn independent base learners and takes the majority/average as the final prediction. Boosting updates weight distribution in each round, and learns base models accordingly, then combines them according to their corresponding accuracy. Different from these two approaches, Stacking [17] learns a high-level model on top of the base models (classifier/regressor). It can be regarded as a meta-learning approach in which the base models are called first-level models and a second-level model is learned from the outputs of the first-level models. A short description of the three methods is introduced below:

- Bagging method generally builds several instances of a black-box estimator from bootstrap replicates of the original training set and then aggregates their individual predictions to form a final prediction. This method is employed as a way to reduce the variance of a base estimator

(e.g., a decision tree) by introducing randomization into its construction process. Random Forest is representative among bagging methods.

- Boosting is a widely used ensemble approach, which can effectively boost a set of weak classifiers to a strong classifier by iteratively adjusting the weight distribution of samples in the training set and learning base classifiers from them. At each round, the weight of misclassified samples is increased and the base classifiers will focus on these more. This is equivalent to inferring classifiers from training data that are sampled from the original data set based on the weight distribution. Gradient Boosting is a mostly used boosting method.

- Stacking involves training a learning algorithm to combine the predictions of several other learning algorithms. First, all of the other algorithms are trained using the available data; then, a combiner algorithm is trained to make a final prediction using all the predictions of the other algorithms as inputs. Stacking typically yields a performance that is better than any single trained models.

*4.1. Random Forest*

As a representative bagging method, RF consists of many individual trees called classification and regression tree (CART), each of which is induced from a bootstrap sample. The CARTs are aggregated to make predictions for a future input; see Figure 4. An underlying assumption is that the base learners are independent. As the trees become more correlated (less independent), the model error tends to increase. Randomization helps to reduce the correlation among decision trees so that the model accuracy is improved. Two kinds of randomness exist in a tree learning process: bootstrap sampling and node splitting. A CART has a binary recursive structure. The tree learning process is actually a recursive partitioning process of the data. The root node corresponds to the whole training data.

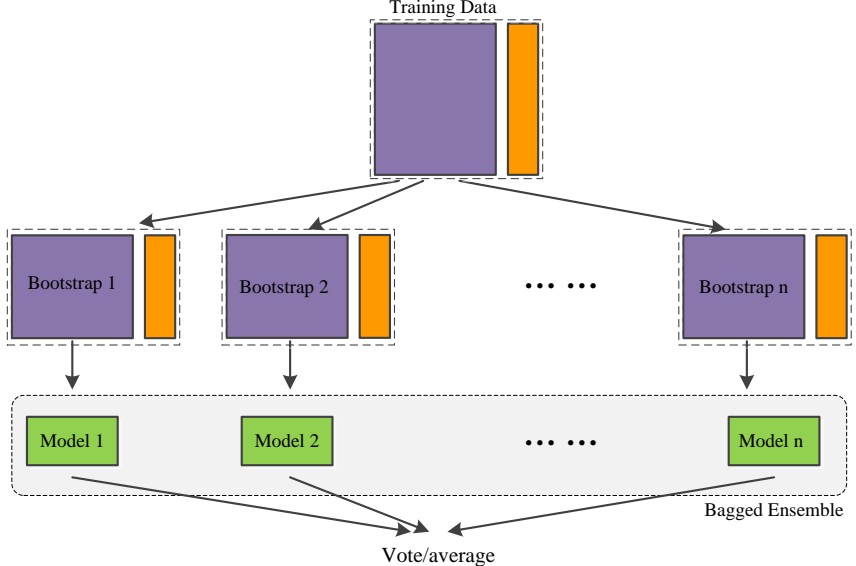

**Figure 4.** Process of bootstrap aggregating.

When a node is split into two child nodes, the data set is simultaneously divided into two subsets according to the splitting point determined by minimizing the sum of squared errors (SSE), i.e., $\min_{i,j}[\sum_{x \in R_1} (y_i - c_1)^2 + \sum_{x \in R_2} (y_i - c_2)^2]$ where $R_1 = \{x | x_i \leq x_{i,j}\}$ is the region that satisfies the condition $x_i \leq x_{i,j}$, $R_2 = \{x | x_i > x_{i,j}\}$ is the region that satisfies $x_i > x_{i,j}$. In addition, $c_m$ is the average of the output values falling into the region, $c_m = E(y | x_i \in R_m), m = 1, 2$. See the illustrative node splitting process in Figure 5.

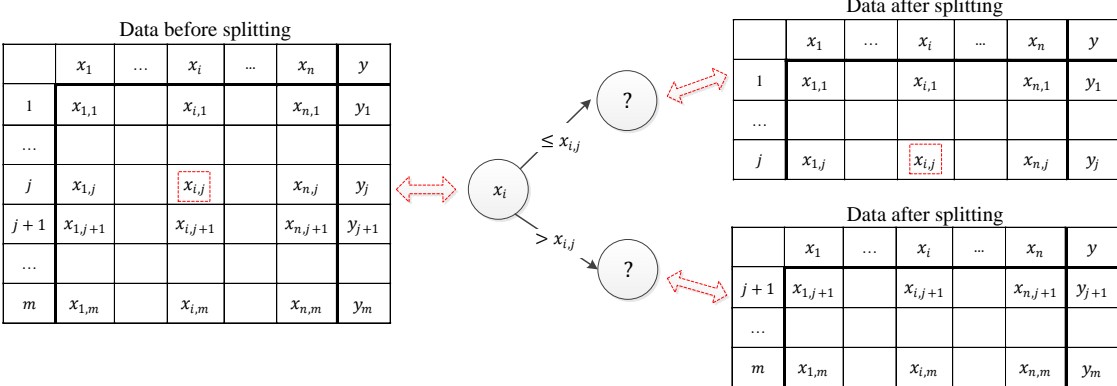

**Figure 5.** An illustrative node splitting process. The symbol '?' means that the variable used to carry out the next splitting needs to be determined.

Similar in principle to RF, another ensemble learning method called Extra-Trees has further randomness in node splitting. Both of them consist of multiple decision trees. In the tree inferring process, both of them use a random subset of the features as the candidate splitting features. The mainly different operations are listed below: (1) the way to prepare the training set for each tree. RF uses a bootstrap replica to train each tree. In contrast, ETs employ the whole training set to train each tree; (2) the way to find the splitting point. In RF, a decision tree firstly finds the best splitting point for each candidate feature, then chooses the best one; however, in an ET, a random splitting point is chosen for each candidate feature; then, the best one is chosen. See Figure 6.

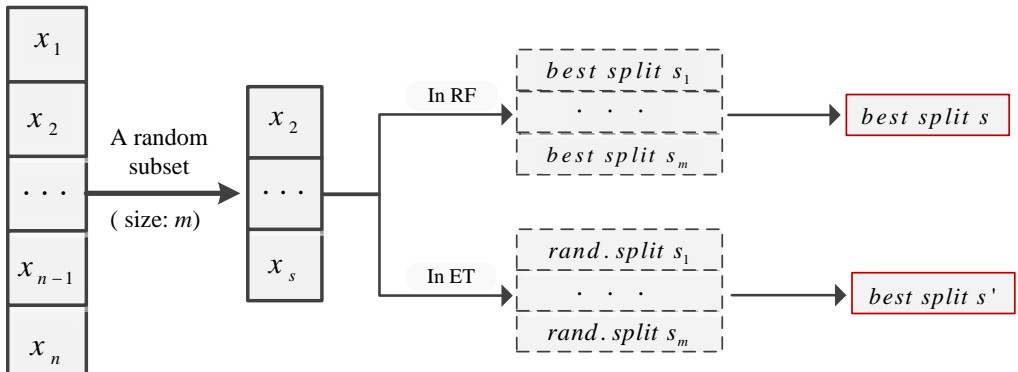

**Figure 6.** Comparison of RF and ETs in node splitting.

*4.2. Gradient Boosting*

Gradient Boosting is a machine learning meta-algorithm. It can be used in conjunction with many other types of learning algorithms to improve their performance. The output of the other learning algorithms ('weak learners') is combined into a weighted sum that represents the final output of the boosted classifier. AdaBoost is adaptive in the sense that subsequent weak learners are tweaked in favor of those instances misclassified by previous classifiers. The individual learners can be weak, but as long as the performance of each one is slightly better than random guessing, the final model can be proven to converge to a strong learner. Weak classifiers can be Decision tree/Decision stump, Neural Network, Logistic regression, or even SVM. To determine the coefficient for each classifier, we initialize the weight distribution of the samples in the dataset by a uniform distribution, and update this distribution by considering the misclassified samples in each loop. Meanwhile, we calculate the coefficient for the current classifier.

Finally, we have a certain number of weak classifiers as well as their weights, then the strong classifier is obtained as the weighted sum of these classifiers,

$$f(x) = \sum_{i=1}^{M} \alpha_t h_t(x) \tag{14}$$

where $\alpha_t$ is the coefficient for the 'weak' classifier $h_t(x)$. For GB, we use all training data, instead of bootstrapping, to build each tree. The first tree is built from an initialized constant model (i.e., a constant value or a one-node tree) [18]. From the second tree, to build each tree, we need to calculate the errors of the former trees (as a weighted sum) on all training points until the stopping criteria (number of trees, error threshold, $\cdots$) are satisfied. See Figure 7.

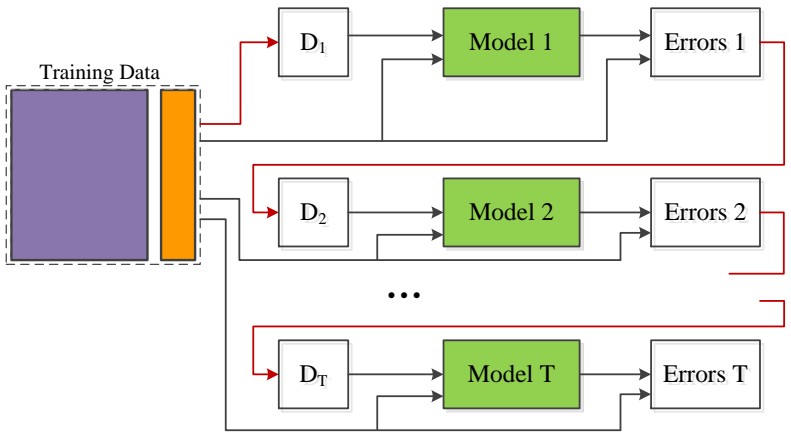

**Figure 7.** Inducing process of Gradient Boosting.

*4.3. Stacking*

Stacking is an ensemble learning technique that learns a meta-learner based on the output of multiple base learners. In a typical implementation of Stacking, a number of first-level individual learners are generated from the training data set by employing different learning algorithms. Those individual learners are then combined by a second-level learner that is called meta-learner. See Figure 8.

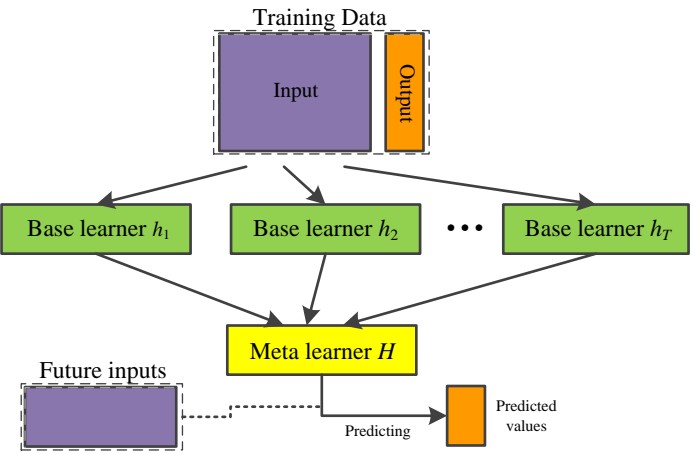

**Figure 8.** General framework of Stacking.

Table 1 shows the Pseudo-code of Stacking, which demands three steps. Firstly, learn first-level (base) learners based on the original training data. There are several ways to learn base learners. We can apply a Bootstrap sampling technique to learn independent learners or adopt the strategy used in Boosting. Secondly, construct a new data set based on the output of base learner. Assume that each example in $D$ is $(\mathbf{x}_i, y_i)$. We construct a corresponding example $(\mathbf{x}'_i, y_i)$ in the new data set, where $\mathbf{x}'_i = h'(h_1(\mathbf{x}_i), h_2(\mathbf{x}_i), \ldots, h_T(\mathbf{x}_i))$. Thirdly, learn a second-level (meta) learner from the newly constructed data. Any learning method could he applied to learn the meta learner.

**Table 1.** Pseudo-code for Stacking [17].

| |
|---|
| **Input:** Training data $D = \{\mathbf{x}_i, y_i\}_{i=1}^m$, $\mathbf{x}_i \in \mathbf{X}$, $y_i \in y$; a Stacking algorithm.<br>**Output:** A meta learner $H$. |
| *Step1:* Induce $T$ base learners, i.e., $h_1, h_2, \ldots, h_T$, from the training set.<br>for $t \leftarrow 1$ to $T$<br>    Learn a base learner $h_t$ from $D$.<br>end for<br>*Step2:* Construct a new dataset $D'$, where $D' = \{(\mathbf{x}'_i, y_i)\}_{i=1}^m$. Here,<br>    $\mathbf{x}'_i = [h_1(\mathbf{x}_i), h_2(\mathbf{x}_i), \ldots, h_T(\mathbf{x}_i)]$.<br>*Step3:* Build a meta-learner $H$ from $D'$. Output $H$. |

Once the meta-learner is generated, for a test example $\mathbf{x}$, its predictions are $h'(h_1(\mathbf{x}), h_2(\mathbf{x}), \ldots, h_T(\mathbf{x}))$, where $(h_1, h_2, \ldots, h_T)$ are base learners and $h'$ is the meta-learner. We can see that Stacking is a general framework. We can plug in different learning approaches or even ensemble approaches to generate first or second level learner. Compared with Bagging and Boosting, Stacking "learns" how to combine the base learner instead of voting. In Table 2, notice that the same data $D$ is used to train base learners and make predictions, which may lead to over-fitting. To avoid this problem, 10-fold cross validations (CVs) is incorporated in stacking.

**Table 2.** Parameters of different models in a 1-DOF case.

| Parameters | RF | GB | ETs |
|---|---|---|---|
| nTrees | 20 | 20 | 20 |
| nFeatures | 3 | 3 | 3 |
| maxFeatures | 2 | 2 | 2 |

## 5. Numerical Examples

This section presents some numerical examples that illustrate the capabilities of the proposed method for estimating first excursion probabilities. In these examples, the proposed approaches are compared with the direct MCS for estimating structural failure probabilities in order to demonstrate the accuracy and efficiency. Firstly, Section 5.1 provides three basic simulations on 1-DOF, 2-DOF and 3-DOF structures. In Section 5.1, a 10-DOF structure is studied as a high-dimensional uncertainty case.

### 5.1. Three Test Examples

The structure is a 1-DOF oscillator, and its parameter values are introduced in Elyes et al. [19], with the nominal values of mass $m = 1.0$ kg, damping coefficient $c = 0.03$ Ns/m, and stiffness $k = 696.4$ N/m. The ground acceleration is modeled by a Kanai–Tajimi (K–T) filter, whose natural frequency and damping ratio are 8 $\pi$rad/s and 0.4 correspondingly. As the input of the K–T filter, the white noise process has the power spectral density (PSD) $S_0 = 0.031$ m$^2$/s$^3$. The failure criterion is $y^* = 0.16$ m. In this example, the uncertainties exist in the three structure properties and take the form of normal distributions with standard deviations (SD) $\sigma_{m_s} = 0.1$ kg, $\sigma_{c_s} = 0.003$ Ns/m,

$\sigma_{k_s} = 69.64$ N/m. Moreover, in the simulations, the random samples of structural properties are censored to avoid the deviation from the mean being larger than five times the SD. The observation time is defined as [0, 20 s] with time step $\Delta t = 0.05$ s. According to the proposed framework, an ensemble model is trained on a training set that takes the samples of structural properties as inputs and CFPs as outputs. Four kinds of ensemble methods are employed to build the models including RF, GB, ETs, and Stacking.

The parameters of the models RF, GB, and ETs are listed in Table 2. 'nTrees' is the number of random trees in the model; 'nFeatures' is the number of features (variables) in the training set; 'maxFeatures' is the number of sampled features in each node split. Notice that the Stacking model is denoted as 'RF&SVM-GB', which means that RF and SVM are employed as base learners and GB the meta-learner. The RF and GB in Stacking have the parameters listed in Table 2; for the meta-model GB, nTrees = 10, maxFeatures = 1. RMSE (Root mean square error) is used to evaluate the performances of the models in fitting CFPs. Based on the proposed framework, the overall failure probability is estimated. See results in Figure 9.

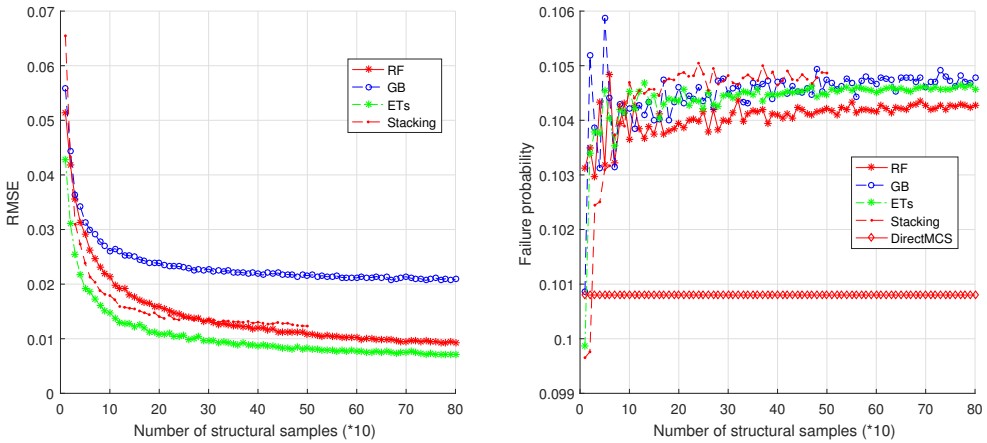

**Figure 9.** Simulation results on a 1-DOF structure.

It can be seen from Figure 9 that the errors of the ensemble models decrease when more samples are added into the training set. The ensemble models RF, ETs, and Stacking have similar RMSEs even though they are slightly different. The GB model always results in higher RMSEs than other three models. By direct MCS, the reference value of failure probability is determined as 0.1008 (see the right part of Figure 9). This reference value is then compared with the result failure probabilities from the four ensemble models. It is found that the four models converge rapidly and converge at $20 \times 10 = 200$ samples. In addition, RF is slightly better than other three methods. One reason is that RF is good at reducing overfitting by randomization of the training process. Actually, the four methods have very close estimations that is about 0.1045. Compared with the reference value, this value is very close to the reference value.

Figure 10 shows the simulation results on a 2-DOF oscillator. The two DOFs have the nominal values of mass $m_1 = m_2 = 4.6$ kg, damping coefficient $c_1 = c_2 = 62$ Ns/m, and stiffness $k_1 = k_2 = 6500$ N/m. The uncertainties exist in all structural properties and take the form of normal distributions with SDs as 10% of the nominal values. The random samples of the uncertain structural properties are truncated in the same way as in the first simulation. The failure criterion is $y_1^* = y_2^* = 0.024$ m. The same ensemble learning methods (with same parameters) are employed as those in a 1-DOF case. It is found that the stacking method outperforms the other three methods in modeling CFPs. The reference value of failure probability is 0.0684 (see the right part of Figure 10). Similar to the evolutions in Figure 9, the four methods converge rapidly to the value 0.0730. Therefore, the four methods still achieve relatively high accuracy in failure probability estimations.

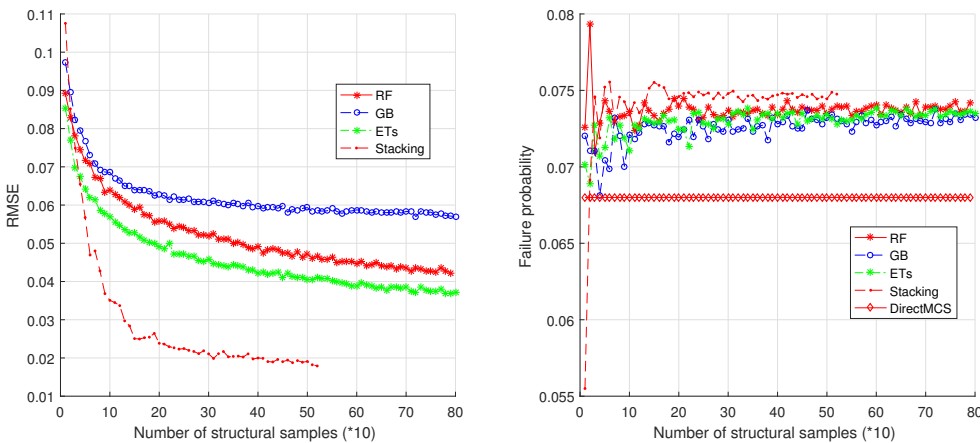

**Figure 10.** Simulation results on a 2-DOF structure.

For the 3-DOF structure, the structural parameters of each DOF and their uncertainties are the same as those in the 2-DOF case. The same ensemble learning methods (with same parameters) are employed as those in the 2-DOF case. The simulation results (see Figure 11) are similar to those on the 2-DOF structure. These results show that the ensemble learning methods give us highly accurate reliability estimations. In addition, from Figures 9–11, it is found that the four ensemble methods converge at the point where the number of structural samples is about 200. In contrast, by direct MCS (see Figure 12), at least 5000 samples are needed to reach a convergence, thus 96% samples are saved.

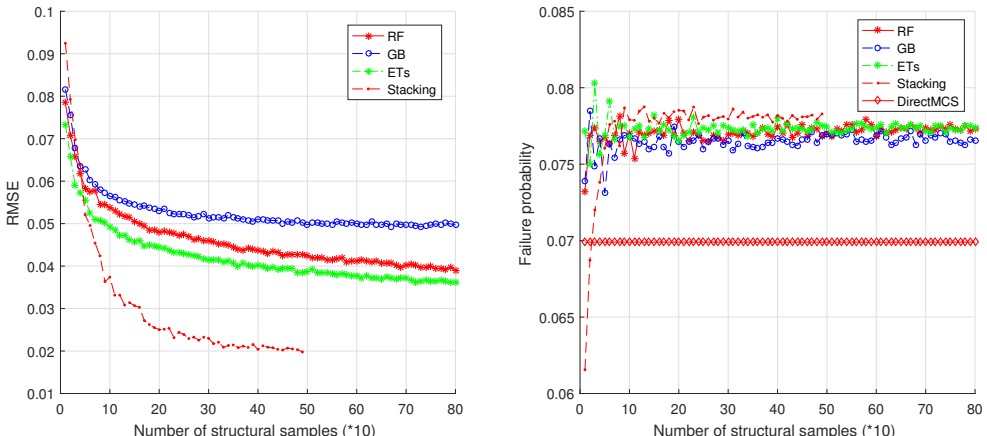

**Figure 11.** Simulation results on a 3-DOF structure.

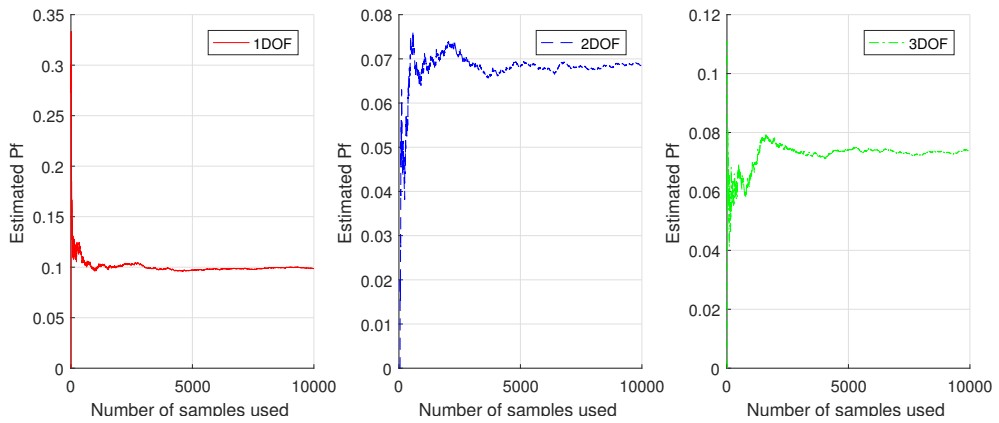

**Figure 12.** Convergence process of direct MCS.

## 5.2. A Benchmark Example: 10-DOF Duffing Oscillator

As a benchmark problem introduced in Schueller [20], the ten-DOF Duffing type oscillator has been widely used by researchers in structural reliability domain. In this study, we focus on the linear random structures under stochastic excitation. The statistical properties of the structural parameters and their constraints are listed in Table 3. Moreover, the Gaussian samples of structural properties are censored in the same way as in Section 5.1. The stochastic excitation $p(t)$ is modeled by a modulated filtered Gaussian white noise:

$$p(t) = \Omega_{1g}^2 v_{f1}(t) + 2\zeta_{1g}\Omega_{1g}v_{f2}(t) - \Omega_{2g}^2 v_{f3}(t) - 2\zeta_{2g}\Omega_{2g}v_{f4}(t), \tag{15}$$

where the state space function with respect to the state vector $v_f(t) = [v_{f1}(t), v_{f2}(t), v_{f3}(t), v_{f4}(t)]^T$ of the filter is

$$\dot{v}_f(t) = \begin{bmatrix} 0 & 1 & 0 & 0 \\ -\Omega_{1g}^2 & -2\zeta_{1g}\Omega_{1g} & 0 & 0 \\ 0 & 0 & 0 & 1 \\ \Omega_{1g}^2 & 2\zeta_{1g}\Omega_{1g} & -\Omega_{2g}^2 & -2\zeta_{2g}\Omega_{2g} \end{bmatrix} v_f(t) + [0, w(t), 0, 0]^T \tag{16}$$

Here, $w(t)$ stands for a modulated Gaussian white noise with auto-correlation function $E(w(t)w(t + \tau)) = I\delta(\tau)h^2(t)$ and $I$ denotes the intensity function of the white noise. $h(t)$ has the following form:

$$h(t) = \begin{cases} 0, & t = 0, \\ t/2, & t \in [0, 2s] \\ 1, & t \in [2s, 10s] \\ exp(-0.1(t - 10)), & t \in [10s, 20s] \end{cases} \tag{17}$$

$\delta(t)$ is the dirac delta function that equals $+\infty$ at $t = 0$ and $0$ at $t \neq 0$. The values $\Omega_{1g} = 15.0$ rad/s, $\zeta_{1g} = 0.8$, $\Omega_{2g} = 0.3$ rad/s, $\zeta_{2g} = 0.995$, and $I = 0.08$ m$^2$/s$^3$ are used to model the filter. The input excitation of the filter is a shot noise that consists of a series of independent normally distributed impulses arranged at each time step. The magnitude of the impulse at time $t_k = k\Delta t$ has mean 0 and standard deviation $h(t)\sqrt{I/\Delta t}$.

**Table 3.** Statistical properties of the structural parameters ($r = SD/\mu$).

| Variables | Mean ($\mu$) | SD | Ratio (r) | Range Scope |
|---|---|---|---|---|
| $m_1, \ldots, m_{10}$ | $10 \times 10^3$ kg | $1.0 \times 10^3$ kg | 0.1 | $\mu \pm 5\mu r$ |
| $k_1, k_2, k_3$ | $40 \times 10^6$ N/m | $4.0 \times 10^6$ N/m | 0.1 | $\mu \pm 5\mu r$ |
| $k_4, k_5, k_6$ | $36 \times 10^6$ N/m | $3.6 \times 10^6$ N/m | 0.1 | $\mu \pm 5\mu r$ |
| $k_7, k_8, k_9, k_{10}$ | $32 \times 10^6$ N/m | $3.2 \times 10^6$ N/m | 0.1 | $\mu \pm 5\mu r$ |
| $\zeta_1, \ldots, \zeta_{10}$ | $620 \times 10^4$ N s/m | $62 \times 10^4$ N s/m | 0.1 | $\mu \pm 5\mu r$ |

The failure criterion is defined by the maximum relative displacements between two consecutive DOFs over the time interval [0.0 s, 20.0 s]. The sampling time step is set as $\Delta t = 0.05$ s. The failure probabilities are calculated for different threshold values listed in Table 4. We are interested in the first excursion probability in which the relative maximum displacement of the first DOF is greater than the threshold 0.057 m and 0.073 m; in addition, the probability that the relative maximum displacement between the 9th DOF and 10th DOF exceeds the threshold 0.013 m and 0.017 m is considered.

**Table 4.** Thresholds of interest to evaluate failure probability [20].

| Failure Defined by | Res. Threshold1 | Res. Threshold2 |
|---|---|---|
| First, DOF | 0.057 m | 0.073 m |
| Tenth DOF | 0.013 m | 0.017 m |

In this simulation, the estimated failure probability values are very small (below $10^{-3}$); therefore, the KL-IS method is employed to calculate the conditional failure probabilities. In applying K-L expansions, the number of K-L terms kept is $n_{KL} = 300$. The parameters of different models are shown in Table 5. The overall failure probability is estimated that is concerned with both structural uncertainties and excitation uncertainties. To make the results more convincing, the estimations by the proposed method are compared with those of other published methods; see Table 6. Notice that '*' means this result comes from the literature. In the simulations, it is found that the ensemble learning based methods all reach a convergence at the point $n\_sample = 500$.

**Table 5.** Parameters of different models.

| Parameters | RF | GB | ETs |
|---|---|---|---|
| nTrees | 30 | 30 | 30 |
| nFeatures | 30 | 30 | 30 |
| maxFeatures | 6 | 6 | 6 |

It is seen from Table 6 that, for high-dimensional reliability problems with respect to linear stochastic dynamics, a number of methods and their variants exist by which this problem can be solved very efficiently when compared to direct MCS. More importantly, the ensemble learning based methods proposed in this research are highly comparable with the already existing methods. It is noticed that the estimated probabilities are very small (e.g., $P_f < 10^{-4}$), which implies that direct MCS is practically not applicable because a very large number of simulations are required. By the comparisons in Table 6, it is found that the number of structural samples analyzed in the proposed methods is very small (e.g., 500), but the finally estimated failure probabilities are of high accuracy. These results reveal that the ensemble learning based methods are powerful in handling high-dimension small probability problems.

<p align="center">**Table 6.** Reliability estimation results from different methods.</p>

| Method | 1st-DOF 0.057$m$ | 1st-DOF 0.073$m$ | 10th-DOF 0.013$m$ | 10th-DOF 0.017$m$ |
|---|---|---|---|---|
| **Standard MCS** | 1.06E-4 | 8.07E-7 | 4.88E-5 | 2.52E-7 |
| *num of samples* | 2.98E+7 | 2.98E+7 | 2.98E+7 | 2.98E+7 |
| **SubsetSim/MCMC** [21] | 1.20E-4 | 1.00E-6 | 6.60E-5 | 4.70E-7 |
| *num of samples* | 1850 | 2750 | 2300 | 2750 |
| **SubsetSim/Hybrid** [21] | 1.10E-4 | 1.10E-6 | 5.90E-5 | 3.20E-7 |
| *num of samples* | 2128 | 3163 | 2645 | 3680 |
| **Complex Modal Ana.** [22] | 1.00E-4 | 9.80E-7 | 6.00E-5 | 4.60E-7 |
| *num of samples* | 300 | 300 | 300 | 300 |
| **Spherical SubsetSim** [23] | 9.20E-5 | 8.80E-7 | 4.60E-5 | 5.30E-7 |
| *num of samples* | 3070 | 4200 | 3250 | 4900 |
| **Line sampling** [24] | 9.80E-5 | 9.70E-7 | 6.00E-5 | 4.60E-7 |
| *num of samples* | 360 | 3600 | 360 | 360 |
| **RF-based** | 7.6E-5 | 1.0E-6 | 4.2E-5 | 1.1E-7 |
| *num of samples* | 500 | 500 | 500 | 500 |
| **GB-based** | 8.48E-5 | 9.15E-7 | 4.24E-5 | 1.06E-7 |
| *num of samples* | 500 | 500 | 500 | 500 |
| **ETs-based** | 8.73E-5 | 9.89E-7 | 4.09E-5 | 1.10E-7 |
| *num of samples* | 500 | 500 | 500 | 500 |
| **Stacking-based** | 1.0E-4 | 9.2E-7 | 4.3E-5 | 1.1E-7 |
| *num of samples* | 500 | 500 | 500 | 500 |

## 6. Conclusions

In this study, an ensemble learning based method is proposed for mechanical reliability assessment. The ensemble learning methods are treated as surrogate models that can be employed to fit the CFPs of the structure. For very small failure probabilities for which direct MCS is practically impossible to realize, an importance sampling technique is employed that constructs the importance sampling density function based on the basic failure events of the mechanical structure. Once the surrogate model is built, the predictions of CFPs can be made on new samples of the structural properties, so that the overall failure probability can be directly estimated. The representative ensemble methods RF, GB, ETs, and Stacking are considered in the proposed method. Numerical simulations are performed on a different number of DOFs structures to examine the performance of the proposed methodology. For low-dimension uncertain structures, i.e., 1-DOF, 2-DOF, and 3-DOF, the four ensemble methods are considered respectively to build the estimation models, and the results reveal that all these models achieve very high accuracies. For high-dimension uncertain structural properties, a benchmark example is introduced from the famous work of G.I. Schueller where different representative reliability methods are introduced and compared. These representative methods are further compared with the proposed method that considers different ensemble models. The simulation results show that, with a very small number of structural samples analyzed, the ensemble leaning based method obtains a significant advantage over direct MCS in terms of efficiency, meanwhile keeping a high accuracy. The comparisons with other published methods make the proposed method highly competitive.

**Author Contributions:** Conceptualization, W.Y., A.S., M.I., and A.-m.Z.; Methodology, W.Y. and A.S.; Investigation, W.Y.; Writing—Original Draft, W.Y.; Writing—Reviewing and Editing, A.S., A.-m.Z., and M.I.; Visualization, W.Y. All authors have read and agreed to the published version of the manuscript.

**Funding:** This research received funds from the China Scholarship Council.

**Conflicts of Interest:** The authors declare no conflict of interest.

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
