# Peer review of "Mechanical Reliability Assessment by Ensemble Learning"

_vehicles, doi:10.3390/vehicles2010007_

Round 1

Reviewer 1 Report

The idea of the article is good but it is not clearly presented. The structure of the paper needs improvement. There are many equations and many variables that the reader needs to go back and front in order to understand. Are all these equations necessary? Some of them could be omitted and the authors could use more references instead to show the evolution of the methods. 

I would suggest a more clear structure. For instance, a section with existing methods and the relative problems of using these, another section with the proposed method, the techniques employed and the advantages, another section with the goodness-of-fit measures, such as RMSE, etc. The RMSE is mentioned in Figure 9 but it is not explained in the text. More measures could be used. Then, a section with the results, the figures, failure probability and reliability results and the physical meaning of these values. Finally, a section for the conclusion. 

The Figure 3 is necessary but needs more explanation. Is it the proposed methodological framework? Sample 1 for training is different from sample 1 for prediction? Then it should be sample A or just use a different name. The process of sampling in both cases is the same? Do you split the data in order to use different data for training and for prediction? 

As future prospect, classification before sampling and then sampling from each class would be an idea.

Minor spell errors such need to be corrected, such as "an indicator" instead of "a indicator".

Author Response

Dear sir/madame,

The attached file is my feedback to your comments. Please see details in this document. Thank you for your patience.

Best wishes

Reviewer 2 Report

In this paper a mechanical reliability estimation by deep learning method is presented, which is an interesting work. The presentation is pretty good, and the mechanical reliability estimation by important sampling and ensemble learning methods are detailed. 

In the numerical example part, more details can be presented for the ensemble learning models, and more result analysis can be given to compare the concerned model.

Author Response

Dear sir/madame,

The attached file is my feedback to your comments. Thank you.

Best wishes

Round 2

Reviewer 1 Report

My comments have been addressed adequately. The paper has been improved and all the components are more clearly described.